# Polymyxin Induces Significant Transcriptomic Perturbations of Cellular Signalling Networks in Human Lung Epithelial Cells

**DOI:** 10.3390/antibiotics11030307

**Published:** 2022-02-24

**Authors:** Mengyao Li, Mohammad A. K. Azad, Maizbha U. Ahmed, Yan Zhu, Jiangning Song, Fanfan Zhou, Hak-Kim Chan, Tony Velkov, Qi Tony Zhou, Jian Li

**Affiliations:** 1Biomedicine Discovery Institute, Infection & Immunity Program and Department of Microbiology, Monash University, Melbourne, VIC 3800, Australia; mengyaoli369@163.com (M.L.); mohammad.azad@monash.edu (M.A.K.A.); yan.zhu@monash.edu (Y.Z.); 2Department of Industrial and Physical Pharmacy, College of Pharmacy, Purdue University, West Lafayette, IN 47907, USA; ahmed156@purdue.edu (M.U.A.); tonyzhou@purdue.edu (Q.T.Z.); 3Biomedicine Discovery Institute and Department of Biochemistry and Molecular Biology, Monash University, Melbourne, VIC 3800, Australia; jiangning.song@monash.edu; 4Monash Centre for Data Science, Monash University, Melbourne, VIC 3800, Australia; 5Sydney Pharmacy School, Faculty of Medicine and Health, The University of Sydney, Sydney, NSW 2006, Australia; fanfan.zhou@sydney.edu.au; 6Advanced Drug Delivery Group, Sydney Pharmacy School, The University of Sydney, Sydney, NSW 2006, Australia; kim.chan@sydney.edu.au; 7Department of Pharmacology and Therapeutic Sciences, University of Melbourne, Melbourne, VIC 3010, Australia; tony.velkov@unimelb.edu.au

**Keywords:** polymyxin, pulmonary toxicity, transcriptomics, lung epithelial cells

## Abstract

Inhaled polymyxins are increasingly used to treat pulmonary infections caused by multidrug-resistant Gram-negative pathogens. We have previously shown that apoptotic pathways, autophagy and oxidative stress are involved in polymyxin-induced toxicity in human lung epithelial cells. In the present study, we employed human lung epithelial cells A549 treated with polymyxin B as a model to elucidate the complex interplay of multiple signalling networks underpinning cellular responses to polymyxin toxicity. Polymyxin B induced toxicity (1.0 mM, 24 h) in A549 cells was assessed by flow cytometry and transcriptomics was performed using microarray. Polymyxin B induced cell death was 19.0 ± 4.2% at 24 h. Differentially expressed genes (DEGs) between the control and polymyxin B treated cells were identified with Student’s *t*-test. Pathway analysis was conducted with KEGG and Reactome and key hub genes related to polymyxin B induced toxicity were examined using the STRING database. In total we identified 899 DEGs (FDR < 0.01), KEGG and Reactome pathway analyses revealed significantly up-regulated genes related to cell cycle, DNA repair and DNA replication. NF-κB and nucleotide-binding oligomerization domain-like receptor (NOD) signalling pathways were identified as markedly down-regulated genes. Network analysis revealed the top 5 hub genes (i.e., degree) affected by polymyxin B treatment were *PLK1*
*(48), CDK20 (46), CCNA2 (42), BUB1 (40) and BUB1B (37).* Overall, perturbations of cell cycle, DNA damage and pro-inflammatory NF-κB and NOD-like receptor signalling pathways play key roles in polymyxin-induced toxicity in human lung epithelial cells. Noting that NOD-like receptor signalling represents a group of key sensors for microorganisms and damage in the lung, understanding the mechanism of polymyxin-induced pulmonary toxicity will facilitate the optimisation of polymyxin inhalation therapy in patients.

## 1. Introduction

Antimicrobial resistance is one of the top significant threats to global public health [1]. Particularly, antibiotic resistant lower respiratory tract infections are a major cause of mortality for all ages and the leading cause of death for children under 5 years [2]. In 2016 lower respiratory tract infections caused approximately 2.38 million deaths worldwide, with over 1 million deaths occurring in adults >70 years old [3]. Increasing multi-drug resistance (MDR) among Gram-negative lung pathogens (e.g., *Acinetobacter baumannii*) is a particularly grave threat in critically ill patients such as those with ventilator-associated pneumonia (VAP) [4]. The polymyxins (polymyxin B and colistin) retain significant activity against many of these MDR pathogens [5,6,7], and are often the last-line therapy for the treatment of these problematic lung infections [8,9].

There have been increasing reports of polymyxin resistance in hospitals worldwide [5,10,11]. Polymyxins are usually administered intravenously for treatment of infections caused by Gram-negative bacteria [12,13,14]. However, the current dosing recommendations for intravenous polymyxins are suboptimal, particularly for the treatment of lung infections where relatively low drug exposure is achieved in epithelial lining fluid (ELF) [14,15,16,17]. Dose-dependent nephrotoxicity that can occur in up to 60% of patients following intravenous administration of the polymyxins [18,19], is the major limiting factor for dose escalation. In contrast, inhaled polymyxins can achieve high drug exposure directly at the site of infection within the lungs while minimising systemic drug exposure [20,21]. Consequently, the use of inhaled polymyxins to treat lung infections has rapidly increased over the last decade [22,23]. However, current inhaled therapy of polymyxins is empirical and pulmonary adverse effects have been frequently reported [23,24,25]. Our previous studies suggest that polymyxin toxicity in human epithelial cells is due to multiple cell death pathways [26,27], the mechanisms underpinning polymyxin-induced pulmonary toxicity remains unclear and no information is available on the interplay of different signalling pathways at the systems level. Given such information is critical for the optimisation of polymyxin inhalation therapy, this study employed transcriptomics to examine polymyxin-induced perturbations of signalling networks in human lung epithelial A549 cells. Our novel findings revealed that polymyxin treatment of human lung epithelial cells at concentrations achievable in the lungs via inhalation therapy induced perturbations in the expression of genes involved in cell cycle, DNA damage and pro-inflammatory NF-κB and NOD-like receptor signalling pathways. 

## 2. Material and Methods

### 2.1. Cell Culture and Treatment

For toxicity assessment, human lung alveolar epithelial A549 cells (American Type Culture Collection, Manassas, MD, USA) were grown and maintained according to the manufacturer instruction [26]. A459 cells (1.2 × 10^5^ cells/well) were seeded in 12-well plates (Corning Costar, Sigma-Aldrich, Australia) and incubated at 37 °C in a humidified atmosphere containing 5% CO_2_ for 24 h prior to experiments. At 24 h, the medium was removed, and fresh medium was added for treatments. Immediately prior to all experiments, solutions of polymyxin B (Beta Pharma, Shanghai, China) were prepared in Milli-Q water (Millipore, North Ryde, Australia) and sterilised using 0.20-μm cellulose acetate syringe filters (Millipore, Bedford, MA, USA). A549 cells were exposed to either 1.0 mM of polymyxin B (treatment group) or Milli-Q water (control group) for 24 h (*n* = 3). Polymyxin B toxicity was measured using Annexin-V-Alexa fluor 488 and propidium iodide (PI) staining with fluorescence activated cell sorting (FACS) [26].

### 2.2. Gene Expression Profiling

For the transcriptomic study, A549 cells were grown in 6-well plates (1 × 10^6^ cells/well) overnight. Cells were exposed for 24 h to either polymyxin B (1.0 mM) or Milli-Q water (control) (*n* = 3). Subsequently, the growth medium was aspirated, and lysis buffer was added to the cell-culture plates. A rubber policeman was used to detach the cells. The lysate was passed through a blunt 20-gauge needle fitted to an RNase-free syringe at least 5 times and RNA was extracted using a RNeasy Plus Mini Kit (QIAGEN, Australia). Quantification of extracted mRNA was performed by the Monash Health Translation Precinct (MHTP) Medical Genomic Facility with Agilent microarray using the Human Gene Expression v3 [28].

### 2.3. Bioinformatic Analysis

Microarray raw intensities were pre-processed to perform background correction, quantile normalisation and log_2_ transformation by Bioconductor package Limma [29]. In cases of multiple probes being mapped to the same gene, the expression value was summarised as the arithmetic mean of the values of the multiple probes (on the log_2_ scale). After data preprocessing, 21,755 genes were obtained for analysis. Differentially expressed genes (DEGs) were identified using Student’s *t*-test with a false discovery rate (FDR) <0.01. All *p* values were adjusted by the Benjamini-Hochberg FDR procedure [30]. The key genes affected in polymyxin B-treated cells were selected by partial least squares-discriminant analysis (PLS-DA). Functional pathway analysis was conducted against the Kyoto Encyclopedia of Genes and Genomes (KEGG) [31] and Reactome hierarchy [32]. The hypergeometric distribution was used to determine whether a pathway was over-presented in the interesting gene list (i.e. DEG list).

The protein-protein interaction (PPI) network was generated with the STRING database [33]. Genes were represented as “nodes”, while the interactions between any two genes/proteins were represented as an “edge”. In the STRING database, interactions between target genes and their functional partners in the network were determined by confidence scores [33]. To ensure binary interactions with highest confidence were generated, interactions with confidence scores ≥ 0.95 were selected. After filtration, 134,240 relationships between 9380 human proteins were collected and networks were visualized by Cytoscape [34].

## 3. Results

Flow cytometry analysis revealed that following exposure to 1.0 mM of polymyxin B for 24 h, 80.9 ± 4.1% of A549 cells were alive and the percentages of early apoptotic, late apoptotic and necrotic cells were low. Therefore, this treatment condition (1.0 mM for 24 h) was applied for the subsequent transcriptomics study. In total, 899 DEGs were identified in response to polymyxin B treatment, comprising 514 up-regulated and 385 down-regulated genes (FDR ≤ 0.01). The top 15 genes affected by polymyxin B were identified using variable influence on projection (VIP) scores (Figure 1). Nine genes (*CD86*, *ALDOC*, *MT1F*, *PCSK9*, *FGFBP1*, *MVD*, *INSIG1*, *RAB33A* and *CYB5B*) were up-regulated and six down-regulated (*BEX2*, *CCL5*, *FAM129A*, *CSTA*, *NUPR1* and *TXNIP*) in polymyxin B-treated cells. The *CD86* gene, which encodes a potent co-stimulator of T and B lymphocyte function, had the highest VIP score (11.8) (Figure 1). The identified DEGs were comprised by cellular processes including metabolic pathways (*CYB5B*, *INSIG1*, *MVD* and *ALDOC*), immune function (*CD86*, *CCL5*, *TXNIP* and *ALDOC*), gene expression (*TXNIP*), and signal transduction (*CD86*, *FGFBP1* and *CCL5*).

### 3.1. Cell Cycle, DNA Replication and DNA Repair Were Up-Regulated by Polymyxin B Treatment

KEGG pathway enrichment analysis showed that the 514 genes up-regulated by polymyxin B treatment were significantly (FDR < 0.05) associated with steroid biosynthesis, cell cycle, the Fanconi anemia pathway, DNA replication, mismatch repair, homologous recombination, terpenoid backbone biosynthesis, mineral absorption, oocyte meiosis and fatty acid biosynthesis (Table 1). Furthermore, Reactome analysis revealed that these genes were significantly enriched in pathways related to cell cycle (100, FDR = 4.8 × 10^−14^), DNA replication (24, FDR = 2.2 × 10^−6^), DNA repair (39, FDR = 2.9 × 10^-5^), and reproduction (18, FDR = 3.4 × 10^−4^). Cell cycle, M phase, cell cycle checkpoints, mitotic prometaphase and chromatin maintenance were affected by polymyxin treatment (Appendix A). Cell cycle was significantly enriched by up-regulated genes in both the KEGG and Reactome databases. Notably, all four phases of the cell cycle, namely G1, S, G2 and M phases were perturbed (Figure 2). Six DEGs (*RPA1*, *H2AFX*, *HIST1H2BJ*, *HIST1H2BE*, *HIST1H2BB* and *HIST4H4*) were shared by five cell cycle-related pathways in Reactome, including the cell cycle, cell cycle checkpoints, chromosome maintenance, meiosis, and cell cycle mitotic pathways (Figure 3).

DNA replication and repair were also significantly perturbed across 23 up-regulated genes (Table 1 and Appendix A). Nine up-regulated genes involved in DNA replication were identified using both KEGG and Reactome, including helicase encoding gene *DNA2*, ligase encoding gene *LIG1*, minichromosome maintenance complex components *MCM4* and *MCM6*, DNA polymerases *POLA2* and *POLD3*, replication factors *RFC2* and *RFC5*, and replication protein *RPA1*. In the significantly enriched KEGG pathways, the Fanconi anemia, mismatch repair and homologous recombination pathways were associated with DNA repair. By mapping DEGs to these three pathways in KEGG and to DNA repair pathways in Reactome, 19 shared genes were identified (Figure 4). The mineral absorption pathway in KEGG was significantly enriched by 8 up-regulated DEGs, comprising 6 metallothionein-encoding genes (*MT1A*, *MT1B*, *MT1E*, *MT1F*, *MT1H* and *MT1X*), the sulphate/anion transporter gene *SLC26A9*, and the zinc transporter gene *ZNT1* (Table 1).

### 3.2. NF-κB and NOD-Like Receptor Signalling Pathways Were Down-Regulated by Polymyxin B Treatment

Analysis with KEGG and Reactome with FDR < 0.05 did not reveal any pathways enriched by the 385 genes down-regulated in response to polymyxin B treatment. However, when the threshold of *p* value < 0.05 was applied, several pathways were significantly enriched in both KEGG and Reactome. In KEGG, aminoacyl-tRNA biosynthesis, amino acid metabolism, NOD-like receptor signalling, apoptosis and NF-κB signalling pathways were affected by polymyxin B treatment (Table 2). In Reactome, pathways responsible for signal transduction were perturbed, including TGF-beta receptor signalling activates SMADs, calcitonin-like ligand receptors, regulation of TNFR1 signalling, NOD1/2 signalling pathway, TNFR1-induced NF-κB signalling pathway, and TNF signalling (Appendix A). Shared by both enrichment methods, NF-κB and NOD-like receptor signalling were the common perturbed pathways and *BIRC3*, *TNFAIP3* and *TRAF1* were the common DEGs in the NF-κB signalling. Four down-regulated genes, namely *BIRC3*, *TNFAIP3*, *CASP4* and *RIPK2* were all mapped to NOD-like receptor signalling.

### 3.3. Key Regulatory Genes of the Cell Cycle Perturbed by Polymyxin B

Using protein-protein interaction (PPI) network analysis, 134,240 relationships between 9380 human proteins were collected with the confidence score of ≥0.95. Next, the DEGs induced by polymyxin B treatment were overlaid. The largest connected component was composed of 142 nodes and 804 edges (Figure 5A). STRING network analysis revealed that the top 5 regulatory hub genes with the highest degrees in the network due to polymyxin B treatment were *PLK1* (48), *CDC20* (46), *CCNA2* (42), *BUB1* (40) and *BUB1B* (37) (Table 3). DEGs in Component A were mostly involved with cell cycle. The death receptor signalling component (Figure 5B) consisted of five down-regulated genes, including *BIRC3* (3) and *TRAF1* (2); DEGs in Component C (Figure 5C) were responsible for cholesterol biosynthesis, with *CYP51A1* (11) having the highest degree in this component. Component D was associated with plasma lipoprotein clearance and signalling by the TGF-beta receptor complex, and Component E with mRNA splicing. DEGs from the smaller components (Figure 5F–I) were associated with GPCR ligand binding, glycolysis, intra-flagellar transport and fatty acyl-CoA biosynthesis. The GPCR ligand binding component was constructed by two up-regulated genes (*HTR7* and *ADM*) and three down-regulated genes (*CALCA*, *ADM2* and *PTHLH*) (Figure 5F). The glycolysis component (Figure 5G) also contained two up-regulated (*PGAM1* and *ENO2*) and three down-regulated genes (*PSAT1*, *BPGM* and *PHGDH*). The fatty acyl-CoA biosynthesis component (Figure 5I) was comprised of four up-regulated genes, namely *ACACA* (2), *FASN* (2), *ACSS2* (1) and *ACSL3* (1).

## 4. Discussion

A major limitation of intravenous polymyxin administration to patients with lung infections is very low drug exposure in the ELF [15,16,17]. The aerosolised direct delivery of polymyxins to the lungs via a nebuliser or as a dry powder circumvents this limitation and as such have a prominent use for the treatment of VAP and cystic fibrosis associated lung infections [35]. Inhalation of polymyxins provides substantial pharmacokinetic/pharmacodynamic (PK/PD) advantages by achieving significantly higher lung exposure than is possible with intravenous administration, while simultaneously minimising systemic exposure and thus nephrotoxicity [17,20]. Indeed, inhaled polymyxins as salvage therapy for the treatment of respiratory tract infections caused by MDR Gram-negative pathogens have shown excellent bacterial killing and high rates of clinical cure and improvement in patients with nosocomial pneumonia [17,36,37].

Despite the potential advantages of aerosolised polymyxins, pulmonary toxicity has nevertheless been reported in patients [26,36]. Our group has previously shown that polymyxin B can cause the activation of caspase-3, 8, and 9, increased expression of the cell surface death receptor FasL, and mitochondrial damage in human lung epithelial A549 cells in a dose- and time-dependent manner [26]. Moreover, the subcellular localisation of polymyxin B in mitochondria, early endosomes and lysosomes of A549 cells indicates the critical role of cellular uptake in its respiratory toxicity [27]. Given that polymyxin-associated toxicities are multifaceted, in the present study we investigated how polymyxin B-induced gene expression profiles in human A549 lung epithelial cells contribute to lung injury. The concentration of polymyxin B (1.0 mM) was chosen considering clinical relevance and resulted in the death of ~19% of A549 cells following 24 h of treatment (Appendix A). This degree of toxicity is in agreement with earlier studies where mild toxicity in A549 cells was observed following similar polymyxin B exposure [26,27]. This sub-lethal concentration of polymyxin B also avoided the activation of non-specific cellular processes in necrotic cells that could degrade mRNA and thus ensured the availability of a sufficient number of viable cells for collection of quality mRNA for analysis [38]. Such information will be critical for the safe delivery of polymyxins to the lungs. The correlative analyses of DEGs, functional enrichment and PPI network revealed not only the key genes which mediated polymyxin-induced cell death, but also regulation of the signalling pathways to trigger cell death. This finding is well-aligned with the substantial increase in late apoptotic cell death following treatment of polymyxin B in the present study (Appendix A).

A major finding of the present study is that polymyxin treatment significantly perturbed cell cycle, DNA repair, NF-κB and NOD-like receptor signalling pathways in A549 cells. Our correlative bioinformatic analysis revealed that several pathways differentially enriched by polymyxin B treatment were involved in the regulation of cellular processes for cell survival. In particular, pathways involved with cellular proliferation, apoptosis and inflammation were enriched, including cell cycle, DNA replication, mismatch repair, and signalling pathways (e.g., NF-κB and NOD-like receptor signalling). Similar enrichment of these pathways was observed following pathway analysis using the Reactome database. We recently reported in human renal proximal tubular HK-2 cells DNA damage and cell cycle arrest were induced by polymyxins [39]. Taken together, our results strongly indicate that DNA damage and cell cycle play key roles in polymyxin-induced toxicity in A549 cells. Further molecular biology and in vivo studies are warranted to confirm these findings.

The Fanconi anaemia pathway repairs DNA inter-strand crosslinks that inhibit DNA replication and transcription [40]. Enrichment of the Fanconi anaemic pathway in the current study indicates a protective response against DNA damage caused by a sub-lethal dose of polymyxin B. Histones are the major protein components involved in the compaction of genomic DNA into higher order structures, with histones H2A with H2B, H3, H4 and H1 (linker) constituting the nucleosome core [41]. Once chromatin is compacted, genes become inaccessible to transcription factors, preventing their expression [42]. The progression of cells from the G1 to S phase, that latter being the phase in which DNA is replicated, increase the rate of histone gene transcription up to 5-fold and results in a 35-fold increase in the histone level [43]. Once the S phase is completed, histone mRNA levels rapidly decrease [43]. We observed significant up-regulation of the histone cluster 1, 2, 4 gene family (*HIST1H1C*, *HIST1H2BB*, *HIST1H2AL*, *HIST1H2BE*, *HIST1H3D*, *HIST1H3E*, *HIST1H3G*, *HIST1H2BJ*, *HIST2H3D* and *HIST4H4*) in polymyxin B treated A549 cells; these results indicate that polymyxin B caused arrest of the cell cycle in the S phase, possibly due to cells inability to repair the polymyxin B induced DNA damage. Furthermore, PPI network analysis revealed key DEGs in cell cycle due to polymyxin B treatment (Figure 5A) and by ranking with their degrees the top five DEGs in the largest component include *PLK1*, *CDC20*, *CCNA2*, *BUB1* and *BUB1B*. *PLK1* encodes a Ser/Thr protein kinase which belongs to the CDC5/Polo subfamily and is highly expressed during mitosis [44]. *PLK1* is also a key regulator of mitosis initiation via control of the CDK1/Cyclin B complex activity which drives the transition of cells from G2 into the M phase [45]. *CDC20* encodes cell-division cycle protein 20, an essential regulator of cell division which activates the anaphase promoting complex (APC/C) responsible for the ubiquitination and degradation of securin and cyclin B ubiquitination, thus promoting the onset of anaphase and mitotic exit [46]. Immunostaining and confocal microscopy studies previously revealed the ubiquitin protein and polymyxin B co-localised in A549 cells after treatment with 0.1mM polymyxin B for 24 h [27]. *CCNA2* encodes cyclin A2, a core cell cycle regulator that participates in controlling both S phase and mitosis by binding to CDK kinases [47]. Multidomain protein kinases BUB1 and BUB1B are the central components of the mitotic checkpoint for spindle assembly [48]. The PPI network analysis results further confirmed that polymyxin B disturbed cell cycle by affecting the expression levels of multiple key regulatory genes.

Polymyxin B also perturbed key regulators of several other important cellular functions, including death receptor signalling, cholesterol biosynthesis, glycolysis and fatty acyl-CoA biosynthesis (Figure 5B,C,G,I). *BIRC3*, which was down-regulated by polymyxin B in the present study (Figure 5B), is involved in the inhibition of apoptosis by binding to tumour necrosis factor receptor-associated factors TRAF1/2 [49]. *CYP51A1* encodes a cytochrome P450 monooxygenase involved in sterol biosynthesis that catalyses the removal of a 14 alpha-methyl group from lanosterol [50]; whereas *MSMO1* encodes methylsterol monooxygenase 1 which catalyses the first step in the removal of the two C-4 methyl groups of 4,4-dimethylzymosterol (Figure 5C) [51]. Cholesterol plays an important role in lung immunity and host reprogramming of cholesterol metabolism is critical to defending pathogens [52,53]. Cholesterol enhances host defense signalling by promoting the assembly and activation of multiple receptors, including Toll-like receptors (TLRs), T cell receptors, B cell receptors and MHCII [54]. Phosphoglycerate mutase 1 (*PGAM1*) catalyses the interconversion of 3-phosphoglycerate and 2-phosphoglycerate in glycolysis (Figure 5G) [55]. During lung injury, hypoxia-inducible factors (HIFs) can transcriptionally induce the expression of multiple genes encoding key enzymes of glycolysis in alveolar epithelial cells [56]. Significantly increased expression of cholesterol biosynthesis and glycolysis suggests that host immunity was activated by polymyxin B. The acyl-CoA synthetase short chain family member 2 (EC:6.2.1.1) encoded by *ACSS2* participates in fatty acyl-CoA biosynthesis by mediating acetyl-CoA synthesis from acetate (Figure 5I) [57]. Although these pathways (Figure 5B–I) were not considered enriched in the KEGG or Reactome analyses, PPI network analysis indicated their perturbation via key regulatory genes by polymyxin B treatment.

NF-κB and NOD-like receptor signalling was significantly enriched by down-regulated genes due to polymyxin B treatment (Table 2). NOD-like receptor is a pattern recognition receptor involved in the recognition of bacterial pathogens in the lungs and chronic sterile inflammatory diseases, such as pneumonia, acute lung injury, chronic obstructive pulmonary disease, pneumoconiosis, acute respiratory distress syndrome and asthma [58]. The best characterised NODs are NOD1 and NOD2; and their stimulation can result in the activation of MAPK and NF-κB [59]. Activated NF-κB leads to the up-regulated transcription and production of inflammatory mediators [59]. Although NF-κB and NOD-like receptor signalling pathways were enriched from down-regulated genes by polymyxin B treatment, several of these genes play a suppressive role in the pathways. cIAP2 (encoded by *BIRC3*) acts as a ubiquitin-protein ligase, negatively regulating NF-κB and NOD-like receptor signalling pathway [60,61]. TNFAIP3 is a terminator of NF-κB signalling, which is a central regulator of immunopathology [62]. Therefore, down-regulation of these negative regulators may suggest potential activation of NF-κB and NOD-like receptor signalling pathways. Perturbations of these pro-inflammatory pathways indicate that polymyxin B treatment might trigger immune responses in human lung epithelial A549 cells; and highlights that a drug-host-pathogen interplay most likely underlies polymyxin therapy in patients with lung infections.

Although *CD86* (FDR = 0.03, FC = 1.7) was not identified as a DEG by a cut-off of FDR < 0.01, it is the gene with the largest fold change in the polymyxin B-treated cells (Figure 1). *CD86* encodes a potent co-stimulator of T and B lymphocyte function [63] and our result indicates that polymyxin B has immune modulatory activity in A549 cells (Figure 1). Activation of T cells requires the interaction of the T-cell receptor with an antigen bound to MHC expressed on antigen-presenting cells, as well as a costimulatory signal [64]. Increased surface expression of the costimulatory molecule CD86 has been reported in monocyte-derived dendritic cells exposed to several other cationic antimicrobial peptides [65]. The over expression of *CD86* in polymyxin B-treated A549 cells suggests the potential role of polymyxins in modulating both innate and acquired immunity by the activation of costimulatory molecules.

Another major finding of the present study is that six metallothionein encoding genes (*MT1A*, *MT1B*, *MT1E*, *MT1F*, *MT1H* and *MT1X*) were significantly up-regulated in response to polymyxin B treatment (Table 1). Metallothioneins are a family of low molecular weight, cysteine-rich proteins that strongly bind and exchange specific metal ions, in particular zinc [66,67]. Metallothioneins plays an important role in homeostasis of essential metals (e.g., zinc and copper), detoxification of toxic metals (e.g., cadmium), and protection against oxidative stress [68,69]. Zinc, cadmium, oxidative stress and/or inflammation can rapidly and substantially up-regulate the transcription of metallothionein genes [70]. Interestingly, a recent study showed significant correlation between the accumulation of polymyxin and increased concentration of intracellular zinc in A549 cells [71]. It has also been demonstrated that polymyxins cause oxidative stress in A549 cells [26]. These results well support the up-regulated metallothionein encoding genes observed in the present study (Appendix A). As stress proteins, metallothioneins might be a hallmark of polymyxin-induced lung toxicity, which requires further investigation.

In conclusion, this study is the first correlative analysis of transcriptomics, pathways and PPI network of human lung epithelial A549 cells in response to polymyxin treatment. Our findings demonstrate a complex interaction of signalling networks related to DNA damage, DNA repair and cell cycle in A549 cells in response to polymyxin B treatment. This study provides key systems pharmacological information for optimizing pulmonary delivery of polymyxins in patients and the ongoing development of safer, new-generation polymyxins.

## Figures and Tables

**Figure 1 antibiotics-11-00307-f001:**
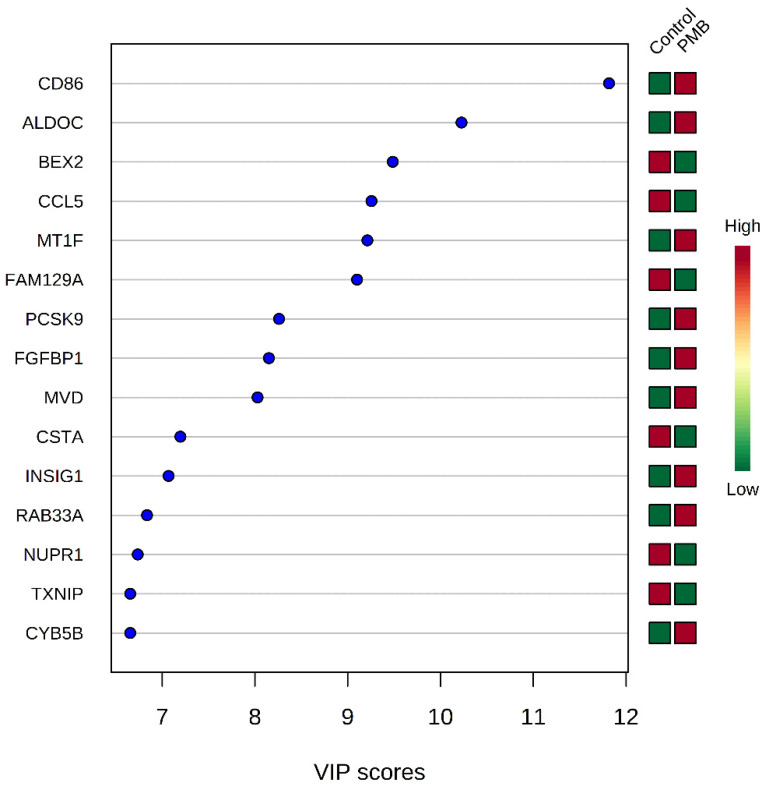
The top 15 genes affected by polymyxin B treatment as identified using variable influence on projection (VIP) scores. Key genes were identified using partial least squares-discriminant analysis (PLS-DA). VIP score and heatmap indicate the relative expression level of the corresponding gene in each group. PMB, polymyxin B.

**Figure 2 antibiotics-11-00307-f002:**
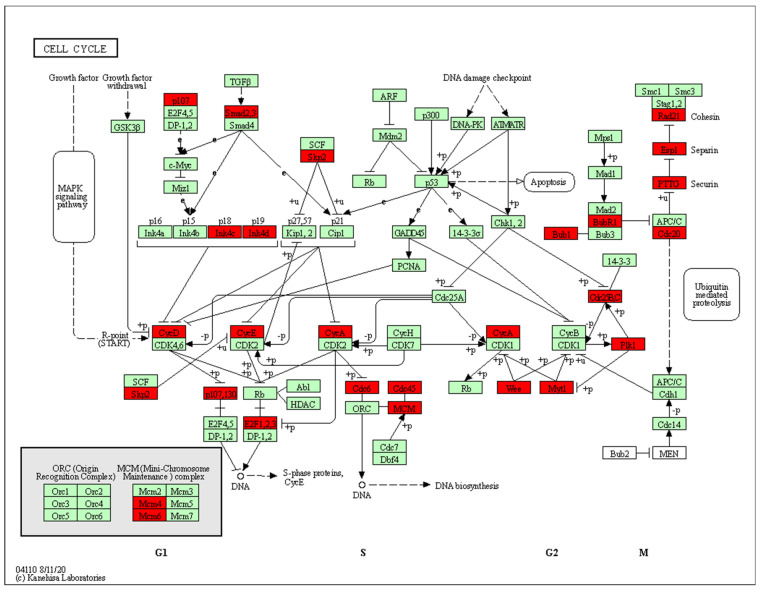
Cell cycle perturbations by polymyxin B. Red boxes present the proteins encoded by up-regulated genes; green boxes present the proteins encoded by down-regulated genes.

**Figure 3 antibiotics-11-00307-f003:**
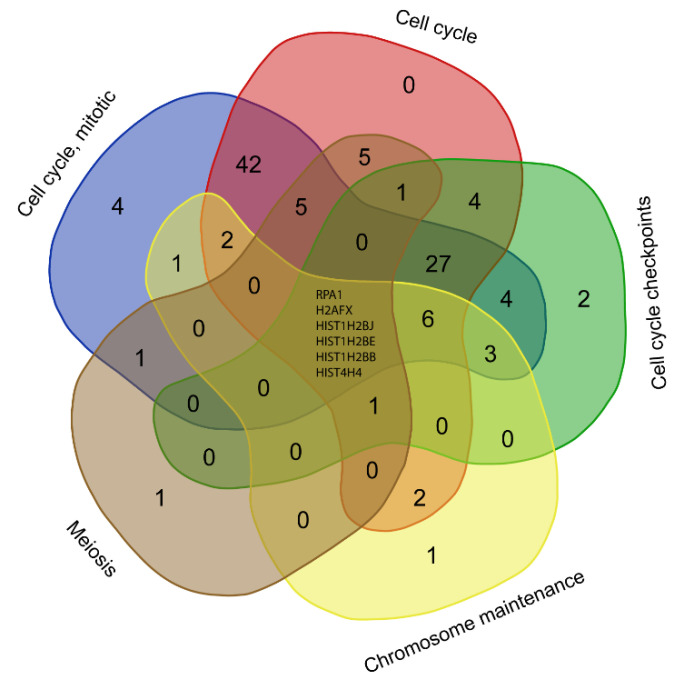
Common DEGs involved in the cell cycle pathways due to polymyxin B treatment.

**Figure 4 antibiotics-11-00307-f004:**
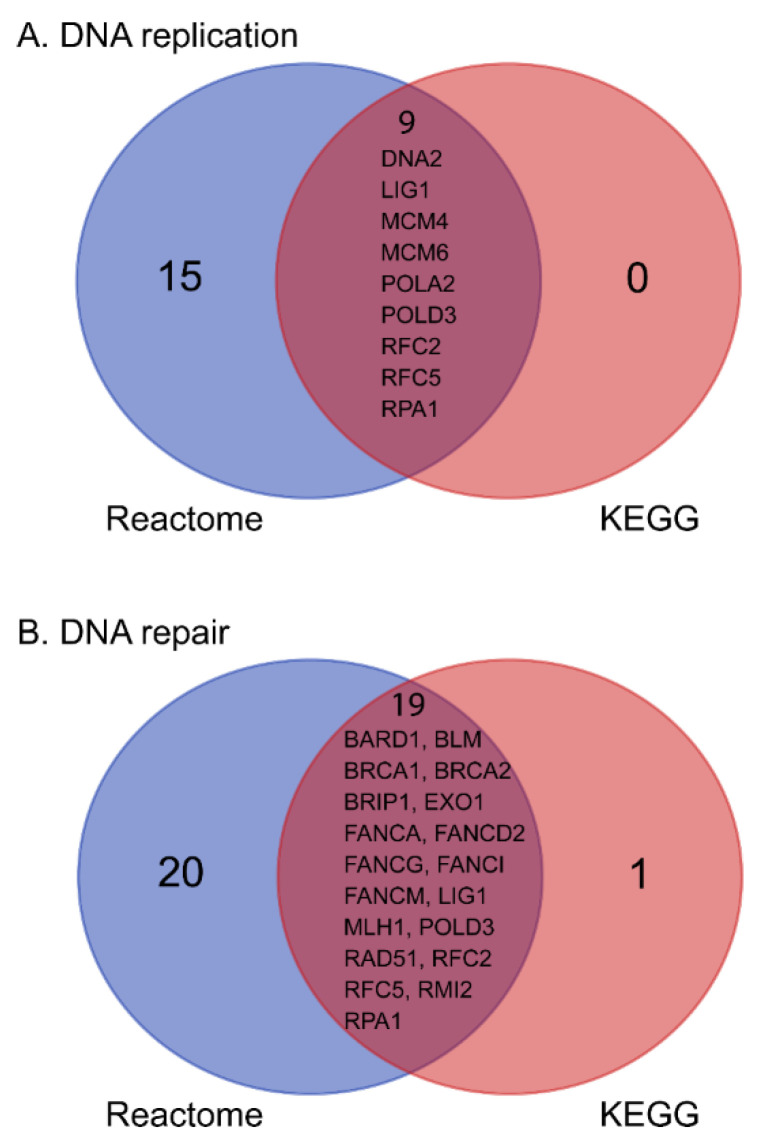
Common up-regulated genes in response to polymyxin treatment shared by Reactome and KEGG. (**A**) Shared up-regulated DEGs in the DNA replication pathway; (**B**) shared up-regulated DEGs in the DNA repair pathway.

**Figure 5 antibiotics-11-00307-f005:**
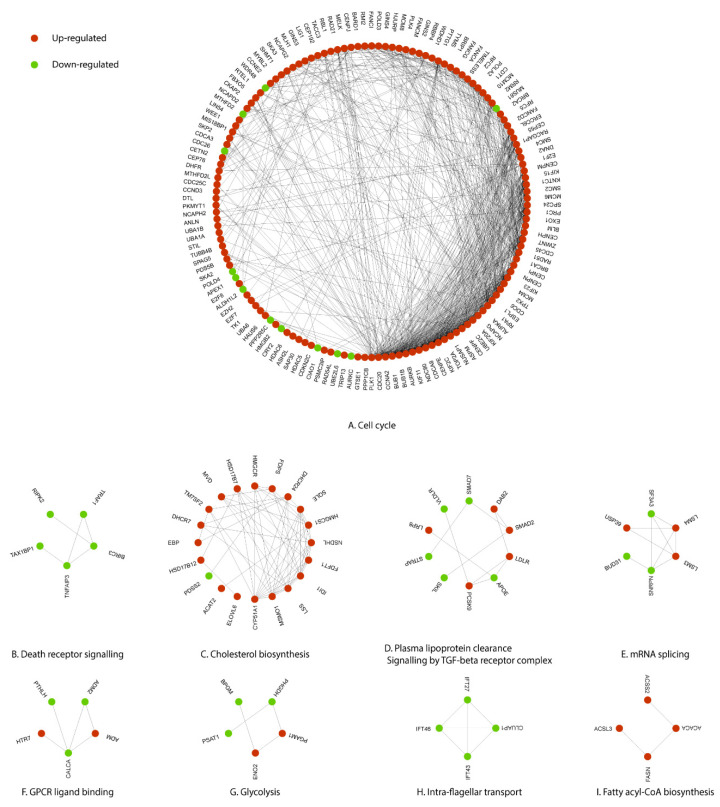
Perturbations of the signalling networks in A549 cells induced by polymyxin B treatment. (**A**) Cell cycle is the largest connected component containing 142 nodes and 804 edges; (**B**) Death receptor signalling component is consisted of 5 down-regulated genes; (**C**) Cholesterol biosynthesis component contains 19 up-regulated genes and 1 down-regulated genes; (**D**) Component D shows 5 up-regulated and 5 down-regulated genes which are related to plasma lipoprotein clearance and signalling by TGF-beta receptor complex; (**E**) mRNA splicing component contains 3 up-regulated and 3 down-regulated genes; both (**F**) GPCR ligand binding and (**G**) Glycolysis components are comprised of 2 up-regulated and 3 down-regulated genes; (**H**) Intra-flagellar transport component contains 4 down-regulated genes, while (**I**) Fatty acyl-COA biosynthesis component contains 4 up-regulated genes. Red and green nodes are up-regulated and down-regulated genes, respectively, following polymyxin B treatment. Interactions between proteins are indicated by lines.

**Table 1 antibiotics-11-00307-t001:** Polymyxin B up-regulated the expression of 514 genes in A549 cells enriched by KEGG pathway analysis (FDR < 0.05). k: number of DEGs in a pathway; m: number of background genes in a pathway; FDR: adjusted hypergeometric *p* value.

KEGG ID	Pathway	k	m	FDR
hsa00100	Steroid biosynthesis	12	19	8.1 × 10^−12^
hsa04110	Cell cycle	23	124	5.2 × 10^−10^
hsa03460	Fanconi anaemia pathway	13	52	4.1 × 10^−7^
hsa03030	DNA replication	9	36	7.7 × 10^−5^
hsa03430	Mismatch repair	7	23	0.2 × 10^−3^
hsa03440	Homologous recombination	9	41	0.2 × 10^−3^
hsa00900	Terpenoid backbone biosynthesis	6	22	0.2 × 10^−2^
hsa04978	Mineral absorption	8	50	0.5 × 10^−2^
hsa04114	Oocyte meiosis	12	119	0.01
hsa00061	Fatty acid biosynthesis	4	13	0.01

**Table 2 antibiotics-11-00307-t002:** Polymyxin B down-regulated the expression of 385 genes in A549 cells enriched by KEGG pathway analysis (*p* < 0.05). k: number of DEGs in a pathway; m: number of background genes in a pathway; *p*: *p* value.

KEGG ID	Pathway	k	m	*p*
hsa00970	Aminoacyl-tRNA biosynthesis	6	44	0.3 × 10^−3^
hsa00260	Glycine, serine and threonine metabolism	5	39	0.1 × 10^−2^
hsa00290	Valine, leucine and isoleucine biosynthesis	2	4	0.3 × 10^−2^
hsa04621	NOD-like receptor signalling pathway	10	170	0.3 × 10^−2^
hsa04210	Apoptosis	8	135	0.8 × 10^−2^
hsa00270	Cysteine and methionine metabolism	4	46	0.02
hsa04064	NF-κB signalling pathway	5	90	0.04

**Table 3 antibiotics-11-00307-t003:** Top 10 perturbed hub genes identified by protein-protein interaction network.

Node	Name	Direction	Degree
PLK1	Polo-like kinase 1	Up	48
CDC20	Cell division cycle 20 homolog	Up	46
CCNA2	Cyclin A2	Up	42
BUB1	Budding uninhibited by benzimidazoles 1 homolog	Up	40
BUB1B	Budding uninhibited by benzimidazoles 1 homolog beta	Up	37
AURKB	Aurora kinase B	Up	36
NDC80	NDC80 homolog, kinetochore complex component	Up	35
KIF11	Kinesin family member 11	Up	35
CENPE	Centromere protein E, 312kDa	Up	33
CDCA8	Cell division cycle associated 8	Up	33

## Data Availability

The microarray data are available from the authors upon reasonable request.

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
