# Peer review of "Polymyxin Induces Significant Transcriptomic Perturbations of Cellular Signalling Networks in Human Lung Epithelial Cells"

_antibiotics, 2022, doi:10.3390/antibiotics11030307_

Round 1

Reviewer 1 Report

The idea to study the gene regulatory system of lung toxicity during exposure with polymyxin B is valid. However, there are certain points that should be addressed, which are listed below:

  • In discussion (line 258-260), the author would like to explain the purpose in these studies how the alterations of gene expression after treatment with polymyxin involve in protection from lung injury. However, the story that explained the association of gene alteration are explained the term of lung toxicity more than the protection. Thus, the author should explain in the detail how alteration of gene expression can become to protect lung injury.
  • The upregulations of gene expression involved in cholesterol and fatty acyl-CoA biosynthesis (line 315-329) are explained only the function in each genes but they did not show the association between alteration of gene expression (cholesterol and fatty acyl-CoA biosynthesis) and lung toxicity. Therefore, the author should mention in the detail.
  • Activation of NF-kB plays an important role for inflammation and immune response. In addition, NOD-like receptor signaling is related with trigger inflammation during bacterial infection. Although both of gene expression showed down-regulation in this study, the author did not mention explain how down-regulation in both gene can trigger immune response (line 337-340). The authors should explain more detail

Reviewer 2 Report

  1. Why authors choose 24h treatment time for Polymyxin? Lower time points with lower treatment concentrations reduce the number of dead cells while making them apoptotic. See the reference below.

https://www.nature.com/articles/srep46541

  1. The reviewer suggests choosing different treatment time periods lower than 24h and analyzing the cells on flow for all time points before going for RNA isolation and transcriptome analysis.
  2. What exactly do authors observe in flow analysis after treatment with Polymyxin? Details are missing in the manuscript. Please justify.
  3. Flow analysis results as shown in supplementary data should be added in the main manuscript while authors should clearly justify in the text, what is what? Why did they select 1.0 mM concentration for treatment for 24h of time?
  4. Is there any differentiation between an apoptotic and necrotic cell in flow experiment observation? Please justify, how this apoptotic and necrotic cell differentiation-related to the final outcomes of affected genes?

Reviewer 3 Report

Dear authors, In section 2.1. Why have you chosen the 1 mM concentration of antibiotic? Have you taken into account different concentrations? Have you considered measuring the impact of antibiotics on cells after more than 24 hours? Would you consider extending the experiments on animal models? It was good to have the supplementary files. Please do check English typing, to eliminate all errors (even small). Did your study have any weakness? Please describe about this in discussion part of the manuscript.
